# Physical Exercise Moderates the Effects of Disability on Depression in People with Multiple Sclerosis during the COVID-19 Outbreak

**DOI:** 10.3390/jcm10061234

**Published:** 2021-03-16

**Authors:** Antonio Carotenuto, Cristiano Scandurra, Teresa Costabile, Luigi Lavorgna, Giovanna Borriello, Lucia Moiola, Matilde Inglese, Francesca Trojsi, Martina Petruzzo, Antonio Ianniello, Agostino Nozzolillo, Maria Cellerino, Giacomo Boffa, Laura Rosa, Alessandro Chiodi, Giuseppe Servillo, Marcello Moccia, Simona Bonavita, Massimo Filippi, Maria Petracca, Vincenzo Brescia Morra, Roberta Lanzillo

**Affiliations:** 1Department of Neurosciences, Reproductive and Odontostomatological Sciences, University of Naples Federico II, 80131 Naples, Italy; carotenuto.antonio87@gmail.com (A.C.); cristiano.scandurra@gmail.com (C.S.); martinapetruzzo@gmail.com (M.P.); servillo@unina.it (G.S.); marcello.moccia@unina.it (M.M.); maria.petracca@unina.it (M.P.); vincenzo.bresciamorra2@unina.it (V.B.M.); roberta.lanzillo@unina.it (R.L.); 2Multiple Sclerosis Centre, II Division of Neurology, University of Campania “Luigi Vanvitelli”, 80131 Naples, Italy; teresa.costabile@gmail.com; 3Department of Advanced Medical and Surgical Sciences, II Clinic of Neurology, University of Campania “Luigi Vanvitelli”, 80131 Naples, Italy; francesca.trojsi@unicampania.it (F.T.); simona.bonavita@unicampania.it (S.B.); 4MS Center, S. Andrea Hospital, Sapienza University, 00189 Rome, Italy; giovanna.borriello@gmail.com (G.B.); antonio.ianniello@uniroma1.it (A.I.); 5Neurology Unit, IRCCS San Raffaele Scientific Institute, 20132 Milan, Italy; moiola.lucia@hsr.it (L.M.); nozzolillo.agostino@hsr.it (A.N.); filippi.massimo@hsr.it (M.F.); 6Department of Neuroscience, Rehabilitation, Ophthalmology, Genetics, Maternal and Child Health (DNOGMI), University of Genoa, 16132 Genoa, Italy; m.inglese@unige.it (M.I.); mariacellerino@hotmail.com (M.C.); boffagcm@gmail.com (G.B.); 7Department of Neurology, Ospedale Policlinico San Martino IRCCS, 16132 Genoa, Italy; 8InterUniversity Center for Research in Neurosciences (CIRN), University of Campania “Luigi Vanvitelli”, 80131 Naples, Italy; rosa-laura@libero.it; 9Intradepartmental Program of Clinical Psychology, Federico II University Hospital, 80131 Naples, Italy; alessandro.chiodi@unina.it; 10Neuroimaging Research Unit, Division of Neuroscience, Institute of Experimental Neurology, IRCCS San Raffaele Scientific Institute, 20132 Milan, Italy; 11Neurophysiology Unit, IRCCS San Raffaele Scientific Institute, 20132 Milan, Italy; 12Vita-Salute San Raffaele University, 20132 Milan, Italy

**Keywords:** COVID-19, multiple sclerosis, depression, anxiety, physical exercise

## Abstract

Physical disability impacts psychosocial wellbeing in people with multiple sclerosis. However, the role of physical activity in this context is still debated. By taking advantage of a previous survey, conducted online from 22 April to 7 May 2020, we performed a post-hoc analysis with the aim to assess the associations between disability, physical exercise, and mental health in multiple sclerosis. We retrieved the following data: (i) sociodemographic information, (ii) changes in lifestyle (including exercise), (iii) physical disability, as measured with the Patient-Determined Disease Steps scale, and (iv) anxiety feelings and depressive symptoms assessed via the items included in the Quality of Life in Neurological Disorders measurement system. Examination of the interaction plot showed that the effect of disability on depression, but not on anxious symptoms, was significant for all levels of physical exercise (low: b = 1.22, 95% C.I. 0.85, 1.58, *p* < 0.001; moderate: b = 0.95, 95% C.I. 0.66, 1.24, *p* < 0.001; and high: b = 0.68, 95% C.I. 0.24, 1.13, *p* = 0.003). Based on these data, we can conclude that disability significantly impacted depression during the COVID-19 pandemic, with physical activity playing a moderating role. Our results suggest that favoring exercise in multiple sclerosis (MS) would ameliorate psychological wellbeing regardless of the level of physical disability.

## 1. Introduction

Multiple sclerosis (MS) is a chronic neurodegenerative disorder characterized by physical, cognitive, and psychological symptoms [1]. Several studies showed disability worsening among people with multiple sclerosis (pwMS) and depression, with physical exercise being related to symptoms of MS [2]. In order to decrease the spread of SARS-CoV-2 during the Coronavirus Disease 2019 (COVID-19), governments decided to send countries into lockdown periods, a restrictive measure mainly focused on the reduction of human contact, which resulted in a drastic decrease in physical activities and an increase in sedentary behaviors. Regardless of COVID-19, people with MS (pwMS) are generally less physically active compared to the general population [3], and these sedentary behavior patterns seem to be increased during the current outbreak [4].

In a previous multicenter study [5], we explored the relationship between mental distress, disability, and coping strategies in an Italian MS sample during the lockdown period, identifying a positive role of active attitudes and an indirect negative role of disability on depressive symptoms. Although this analysis exercise emerged as one of the contributing factors to active attitudes, suggesting that favoring exercise could have a beneficial impact on depression, this might prove difficult in patients with high disability levels. Indeed, although exercise seems to have a protective effect against depressive symptoms, people with a greater level of physical disability may not get the same benefits as people with low disability levels. Additionally, it is still unclear if exercise could exert a positive impact on other aspects of mental health, such as anxiety disorders.

In order to further investigate these aspects, we conducted a post-hoc analysis that aimed to assess the associations between disability, physical exercise, and mental health in pwMS. We anticipated that higher disability correlates with worse mental health outcomes, and physical exercise would moderate this relationship (Figure 1).

## 2. Methods

### 2.1. Procedures and Participants

Data from 497 patients with MS were collected via online survey, developed through the European Commission’s official survey management tool (https://ec.europa.eu/eusurvey), from 22 April to 7 May 2020. The survey was disseminated from 22 April to 7 May 2020 through SMsocialnetwork.com, an online social network addressed to pwMS, and the Facebook page of the MS Center of the University of Naples Federico II. Further details about the survey content can be found in Costabile et al. [5].

For the current analysis, we extrapolated the following data: (i) sociodemographic and clinical information (i.e., age, gender, and MS duration), (ii) changes in lifestyle (including exercise), (iii) physical disability, as measured with the Patient-Determined Disease Steps (PDDS) scale, and (iv) anxiety feelings and depressive symptoms assessed via the items included in the Quality of Life in Neurological Disorders (Neuro-QoL) measurement system. The study was approved by the Carlo Romano ethics committee of the University of Naples Federico II (n.160/20) and was performed in accordance with the Declaration of Helsinki, UE regulations 2016/679 and 2018/1725. All patients and controls gave informed consent before participating in the online survey.

### 2.2. Statistical Analyses

Statistical analyses were performed with SPSS version 26, setting the level of significance at 0.05.

Correlations between variables were calculated using the Pearson’s coefficient. Two hierarchical multiple regression analyses were performed to test the hypothesis that anxiety and depression are functions of disability and, more specifically, that physical exercise would moderate the relationship between disability and mental health. For each model, in the first step, disability and physical exercise were included as independent variables. Then, an interaction term was created between disability and physical exercise, centering both variables and including the interaction in step 2 of the regression model, assessing if such interaction would add explained variance to the model. Finally, to evaluate the conditional effect of disability on anxiety and depression at different values of physical exercise (−1 standard deviation (SD), mean, +1SD), the PROCESS Macro for SPSS was used, applying Model 1 with 5000 bias-corrected bootstrap samples [6]. This analysis was controlled for age, gender, and MS duration.

## 3. Results

Participants ranged in age from 19 to 73 years (M = 42.41; SD = 10.72). In regard to gender identity, 351 (70.6%) were women and 146 (29.4%) were men. The distribution of disease duration was less than 2 years for 43 participants (8.7%), between 2 and 5 years for 81 participants (16.3%), between 5 and 10 years for 105 participants (21.5%), between 10 and 15 years for 164 participants (33%), and more than 20 years for 102 participants (20.5%).

Means, standard deviations, and bivariate correlations between disability, physical exercise, and anxious symptoms are shown in Table 1. The results highlighted a negative correlation between disability and physical exercise as well as between physical exercise and mental health outcomes, and a positive correlation between disability and mental health outcomes.

The hierarchical multiple regression analyses showed that disability and physical exercise accounted for a significant amount of variance in anxiety (R^2^ = 0.07, F (2, 494) = 19.84, *p* < 0.001) and depression (R^2^ = 0.11, F (2, 494) = 29.55, *p* < 0.001). Specifically, both disability and physical exercise increased the likelihood of reporting anxiety (b = 0.15, *p* = 0.001; b = −0.21, *p* < 0.001) and depression (b = 0.25, *p* < 0.001; b = −0.17, *p* < 0.001).

The inclusion of the interaction term between disability and physical exercise increased the explained variance of depression (ΔR^2^ = 0.01, ΔF (3, 493) = 4.85, *p* = 0.002; b = −0.13, *p* = 0.003), but not anxiety (ΔR^2^ = 0.002, ΔF (3, 493) = 1.26, *p* = 0.261; b = −0.07, *p* = 0.261). Thus, interaction plot was assessed only for depression.

Examination of the interaction plot showed that the effect of disability on depression was significant for all levels of physical exercise (low: b = 1.20, 95% C.I. 0.83,1.57, *p* < 0.001; moderate: b = 0.93, 95% C.I. 0.64, 1.23, *p* < 0.001; and high: b = 0.68, 95% C.I. 0.24, 1.13, *p* = 0.003) (Figure 2), but was stronger for those who exercised less. As the effect of disability on depressive symptoms decreases as much as physical exercise increases, this result demonstrates that practicing physical exercise may protect pwMS from the effects of disability on negative mental health outcomes. However, this is particularly valid for women (b = −2.01, *p* = 0.002) and younger (b = −0.08, *p* = 0.006) participants, regardless of the SM duration (b = −0.01, *p* = 0.967).

## 4. Discussion

Our results suggest that, during the COVID-19 outbreak, physical exercise moderated the impact of disability on depression but not on anxiety in pwMS. Specifically, we observed that, although the time devoted to physical activity decreased with increased disability, exercising even in case of high levels of disability protects pwMS from depressive outcomes.

PwMS are typically treated with immunosuppressives, placing them at-risk of infection from viruses [7]. In addition, pre-existing neuropsychiatric symptoms in pwMS put MS patients at higher risk of impaired psychosocial functioning, ultimately resulting in feelings of stress, helplessness, depression, and fear of becoming ill and dying [8]. Therefore, it is crucial to evaluate factors impacting on pwMS’s mental health to prevent disability worsening and to implement tailored treatment to tackle stressful situations. 

As expected, the initial analysis indicated that physical exercise, disability, and depressive and anxious symptoms are strongly correlated with each other, and that both disability and psychological symptoms have an overall negative impact on quality of life [9]. Although physical disability may induce anxious and depressive symptoms per se, the mitigation effect of physical activity was not completely surprising. Generally speaking, physical activity is associated with a better quality of life through an indirect pathway, as disability, anxious and depressive symptoms, fatigue, and pain contribute to quality of life as well [9,10,11]. Recently, Sadeghi Bahmani et al. showed that in patients with MS, participation in regular exercise impacted positively on their objective and subjective sleep, depression, paresthesia, fatigue, and cognitive performance [12].

Differently from previous studies, we hypothesized that physical exercise moderates the effect of disability on psychosocial functioning. While the effect of physical activity on the association between disease-related disability and depression is straight forward, the null effect of physical activity on anxiety deserves further investigations.

A meta-meta-analysis evaluating the impact of physical activity on anxiety in adult subjects reported that physical activity does not ameliorate anxious symptoms in adult people (Rebar, Stanton et al. 2015). Previous studies exploring the association between physical exercise and anxiety in MS were not consistent [13,14], or found small effects of exercise on anxiety in MS [15].

The way physical exercise impacted on anxious and depressive feelings may depend on several causes. Firstly, as emerged in Costabile, Carotenuto et al. [5], pwMS and controls differed for depression severity, but not for the anxious one, with the former being significantly higher in the neurological group. However, anxiety and depressive disorder are often interrelated, since the latter could be a consequence of the former. In particular, anxious symptoms, such as avoidance or the presumed desire to control external events, can turn into behavioral repertoire narrowing and helplessness, which are mostly depressive features. Thus, this interconnection led us to focus on two issues.

The first one concerns the moment we run the online survey, as data were collected close to “phase two”, in which the lockdown restrictions began to loosen, and a gradual resumption of economic activities was expected.

This could have mitigated the level of state anxiety, which may have been greater in the peak stage of the COVID-19 outbreak, as seen in Wang et al. [16] and Demir, Bilek and Balgetir [17].

The second point concerns the meaning that physical exercise has from a psychological point of view. Specifically, it represents a form of behavioral activation, a psychotherapeutic intervention that helps people with depression to functionally cope with negativity, to promote mastery, and to re-build personal short, medium, and long-term life goals [18].

Therefore, physical exercise may mitigate the effect of disability on depression, boosting the adoption of positive attitudes toward difficulties, such as physical disability or the COVID pandemic.

Results of the current study should be read in light of some limitations. First, being an online, anonymous, and self-reported survey, we could not ask participants for their MS phenotypes (e.g., relapsing-remitting or progressive multiple sclerosis). Future studies should control results by analyzing potential differences among groups. In addition, we do acknowledge that the effect in this report is not as large as expected. This may be probably due to the nature of tools used to assess mental health, and the principal component analysis method that may somehow not have captured the whole magnitude of mental health change. Additionally, study duration may have impacted our results. COVID pandemic underwent fluctuations over time and, hence, not limiting our analysis to a restricted time frame could have captured the impact of the pandemic on mental health in periods with different stress extents.

## 5. Conclusions

Our study demonstrated that disability influenced depression in pwMS during thepandemic, through the moderating role of physical activity. Although our results can only be interpreted in the context of the extraordinary situation people are facing, future studies may explore the moderation effect of physical activity on depression and anxiety in conventional contexts, in order to implement tailored rehabilitative programs aimed at ameliorating both physical disability and psychological wellbeing.

## Figures and Tables

**Figure 1 jcm-10-01234-f001:**
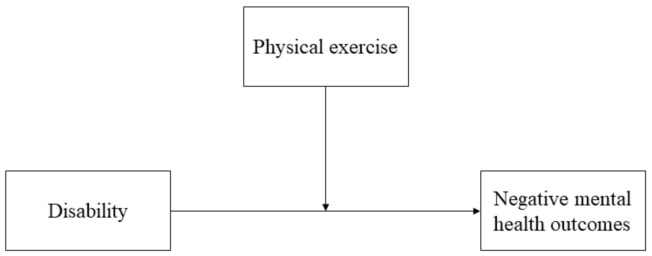
The hypothesized moderation model.

**Figure 2 jcm-10-01234-f002:**
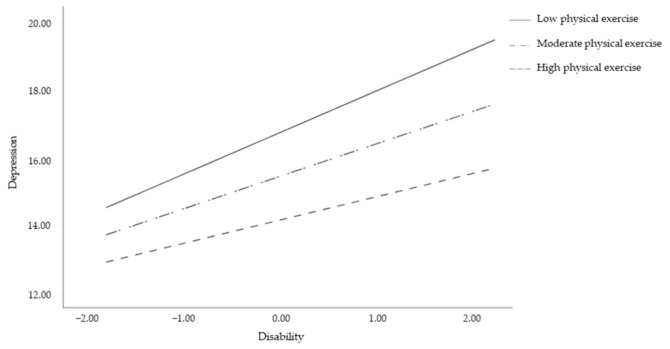
Interaction effect of disability by physical exercise on depression.

**Table 1 jcm-10-01234-t001:** Correlations between disability, physical exercise, and mental health.

	1	2	3	4	M	SD
1. Disability	1				1.83	2.23
2. Physical exercise	−0.16 ***	1			0.77	0.77
3. Anxiety	0.18 ***	−0.23 ***	1		19.61	7.25
4. Depression	0.28 ***	−0.21 ***	0.76 ***	1	15.54	6.99

Note. M = Mean; SD = Standard Deviation. *** *p* < 0.001.

## Data Availability

Anonymizes data will be made available upon reasonable request to the corresponding author.

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
