# Peer review of "Physical Exercise Moderates the Effects of Disability on Depression in People with Multiple Sclerosis during the COVID-19 Outbreak"

_jcm, 2021, doi:10.3390/jcm10061234_

Round 1
Reviewer 1 Report
The authors have designed a study to investigate the associations between disability, physical exercise, anxiety and depression in MS. This is a short duration questionnaire based study in large sample cohort of MS patients. The findings are interesting, but the bivariate correlations studying this are small (though statistically significant). Therefore this should be reflected in the conclusions.
My other suggestions for improvement are:
1) the cohort demographics (age, MS subtype, disease duration) are presented in the results; also were age and sociodemographic factors controlled for in the model?
2) a discussion on the limitations of the study i.e. the study duration for example would be good.
Reviewer 2 Report
Article by Carotenuto et al entitled “Physical Exercise Moderates the Effects of Disability on Depression in People with Multiple Sclerosis During the Covid-19 Outbreak” described the linkage between physical disability/exercise with MS during COVID. I have enjoyed reading this. While I see there is a merit of this work, in my opinion, it needs some revision.
-Did the author found/investigated the impact on different phenotypes of MS such as in RRMS, PPMS, SPMS, and so forth; gender effect, or age-depended changes.
-Please demonstrate lockdown or other terminologies similar to this because now it is COVID so everyone understands this but after several years if anyone reads this who may not have an idea about the COVID situation or these terminologies in relation to MS. So, make it general.
-Moreover, provide a brief discussion about MS and COVID
-At the end of the introduction please provide the result briefly
-Please provide a schematic of how data were acquired and analyzed with n number-male female, followed by key results
-Authors found physical exercise moderated the impact of disability on depression but not anxiety. Please discuss why? In general, anxiety and depression is contradictory to each other. Depression may link with the reduction of Neurotransmitters (e.g. dopamine) or changes in socioeconomic changes (e.g. COVID) whereas elevation of neurotransmitters is linked with anxiety or changes in socioeconomic status. It is a kind of a vicious circle. It would be beneficial if authors discuss how these two are linked with COVID and MS as a whole individually and link with one another.
-Authors discussed differential results between studies may depend upon the tools used to assess anxiety-please provide an explanation with references.
-Please provide the demographic data
-Please provide the limitation of this study and future recommendation
Round 2
Reviewer 1 Report
Much better, no further changes.